

# Unravelling agronomic performance and genetic diversity of newly developed maize inbred lines for arid conditions

Abdallah A. Hassanin[1], Areej S. Jalal[2], Ehab M. Mahdy[3], Hend Mandour[1], Elsayed Mansour[4], Mohamed M. Kamara[5], Mohammed O. Alshaharni[6], Eman Fayad[7], Mohammed Alqurashi[7], Maha Aljabri[8], Thorya Abdulrahman Fallatah[9], Diaa Abd El-Moneim[10] and Ahmed S. Eldomiaty[1]

[1] Genetics Department, Faculty of Agriculture, Zagazig University, Zagazig, Egypt
[2] Department of Biology, College of Science, Princess Nourah bint Abdulrahman University, Riyadh, Saudi Arabia
[3] National Gene Bank (NGB), Agricultural Research Centre (ARC), Giza, Egypt
[4] Department of Crop Science, Faculty of Agriculture, Zagazig University, Zagazig, Egypt
[5] Department of Agronomy, Faculty of Agriculture, Kafrelsheikh University, Kafr El-Sheikh, Egypt
[6] Biology Department, College of Science, King Khalid University, Abha, Saudi Arabia
[7] Department of Biotechnology, College of Sciences, Taif University, Taif, Saudi Arabia
[8] Department of Biology, Faculty of Science, Umm Al-Qura University, Makkah, Saudi Arabia
[9] Department of Biological Science, College of Science, University of Jeddah, Jeddah, Saudi Arabia
[10] Department of Plant Production (Genetic Branch), Faculty of Environmental and Agricultural Sciences, Arish University, El-Arish, Egypt

Corresponding author
Abdallah A. Hassanin, asafan@zu.edu.eg

## ABSTRACT

Investigating genetic diversity of maize inbred lines is crucial for enhancing breeding for higher yields, resilience, and ensuring sustainable maize production amidst climate change and the rapidly growing global population. This study aimed to evaluate the phenological attributes, plant stature, ear characteristics, and grain yield of 14 newly developed Egyptian maize inbred lines across three growing seasons under arid conditions in Egypt. Furthermore, the assessment of the genetic diversity among these lines using three molecular marker techniques: start codon targeted (SCoT), conserved DNA-derived polymorphism (CDDP), sequence-related amplified polymorphism (SRAP). Field evaluation revealed considerable variations in phenological characters including days to tasseling and silking. The earliest maturing lines such as IKA22 demonstrate potential suitability for short growing seasons. Plant and ear heights varied considerably. Taller lines (such as LCM54 and RA28C) potentially offered greater photosynthetic capacity, while shorter lines (such as IKA22, B17AB, and SSK36) may have exhibited improved lodging resistance, especially under adverse weather conditions. Ear diameter and length fluctuated across seasons, with ZBM40A, RA28C, DKCA2, and LZAM7B exhibiting favorable ear length and diameter. The number of rows per ear and kernels per row varied across seasons, with LZAM7B, RA28C and ZBM40A demonstrating the best performance. Grain yield per plant also revealed seasonal variation, with RA28C, ZBM40A, DKCA2, and LZAM7B showing higher yields. The assessed lines were grouped into six categories based on yield-related traits with RA28C demonstrating the best overall performance, followed by ZBM40A, DKCA2, and LZAM7B. Principal component analysis (PCA) identified associations between these lines and yield-related traits, emphasizing their potential as the most promising

candidates for improving maize yield. Moreover, heatmap clustering confirmed the genetic potential and divergence of these lines. PCA demonstrated robust associations between grain yield and critical traits such as rows per ear, ear diameter, and kernels per row, emphasizing their importance for indirect selection. Molecular analysis amplified a total of 95 loci, of which 74 were polymorphic, reflecting substantial genetic variability. The percentage of polymorphism ranged from 0% to 100%, averaging 72.46%. Genetic distance analysis revealed a range of similarities and dissimilarities among these lines. The closest genetic similarity was observed between LZAM7B and ZBM40A, while the greatest divergence was found between LZAM7B and DKCA2. These findings offer valuable insights into the genetic potential of the studied inbred lines, laying a strong foundation for developing resilient, high-yielding maize hybrids for arid and semi-arid environments. Integrating phenotypic and molecular data provides valuable insights for selecting suitable inbred lines in hybrid maize breeding.

## INTRODUCTION

Maize (*Zea mays* L.) is a fundamental cereal crop in global agriculture as a primary source of food, feed, and industrial raw materials (*Amanjyoti et al., 2024*). Its high productivity across diverse agroecological regions has made it a crucial component of sustainable agricultural systems (*Erenstein et al., 2022*). However, the growing global population, changing climatic conditions, and limited natural resources pose substantial challenges to maize production (*Li & Tian, 2024*). These challenges necessitate developing high-yielding, resilient maize hybrids that can withstand fluctuating environmental conditions.

Genetic diversity in maize germplasm is essential for breeding high-yielding hybrids (*Hou et al., 2024*; *Kamara et al., 2022*). It provides a basis for selection and enhances maize's adaptability and resilience to a wide range of environmental conditions (*Omar et al., 2022*). A diverse genetic pool enhances the probability of identifying unique traits or improved nutritional quality, which is essential for meeting modern agricultural demands (*Prasanna et al., 2020*). Maintaining genetic variability is vital for continuous improvement of crops, particularly in response to climate change (*Mansour et al., 2018*; *Snowdon et al., 2021*). Exploring the genetic diversity among maize genotypes is vital for successful breeding programs (*Kamara et al., 2021*). Identifying genetically diverse individuals and populations enables breeders to create diverse breeding populations, increase the likelihood of identifying superior genotypes, and enhance the genetic gain of breeding programs (*Sanchez et al., 2023*). Additionally, genetic diversity information can be used to generate core collections of germplasm that represent the broader genetic variability of a larger collection, thereby promoting efficient conservation and utilization of germplasm (*Ruperao, 2024*).

Identifying inbred lines with superior agronomic traits is essential for developing high-yielding, adaptable hybrids (*Luz et al., 2024*). Among the key traits influencing maize

productivity are phenological attributes (*e.g.*, days to tasseling and silking), plant stature (*e.g.*, ear height and plant height), ear characteristics (*e.g.*, ear length, ear diameter, and kernel traits), and grain yield. Early maturity is particularly critical for regions with short growing seasons or limited water availability. Moreover, characteristics such as ear height and plant height influence lodging resistance, which is a crucial factor in ensuring stable yields under adverse conditions (*Niu et al., 2022*). Ear characteristics including ear diameter, length, and kernel traits, directly correlate with grain yield and are key indicators of a line potential for hybrid development (*Mueller, Messina & Vyn, 2019*). Hence, evaluating the agronomic performance of inbred lines is critical for identifying high-performing and stable genotypes. In particular, multi-season evaluations provide essential insights, allowing breeders to identify lines that are genetically stable and consistently exhibit superior traits across different growing conditions (*Appunu et al., 2024*).

Molecular markers have revolutionized the assessment of genetic diversity, providing precise and reliable insights into genetic relationships among individuals and populations (*Bidyananda et al., 2024*; *Ramesh et al., 2020*). Recently, various molecular marker techniques have been employed to study genetic diversity in maize, including simple sequence repeats (SSRs), amplified fragment length polymorphism (AFLP), and single nucleotide polymorphisms (SNPs) (*Antoni Rafalski & Tingey, 1993*; *Prasanna, 2012*; *Venadan et al., 2024*). Additionally, start codon targeted (SCoT), conserved DNA-derived polymorphism (CDDP), and sequence-related amplified polymorphism (SRAP) have been widely used in genetic diversity studies due to their simplicity, reproducibility, and ability to detect polymorphisms across the genome (*Rizk et al., 2024*; *Tahir et al., 2023*). These markers have been instrumental in unravelling the genetic structure of maize populations, identifying genetic markers associated with important agronomic traits, and facilitating marker-assisted selection (MAS) in breeding programs (*Sobiech et al., 2022*).

Maize breeding in arid regions necessitates germplasm with genetic diversity that balances potential and resilience to environmental stress. While traditional phenotyping remains vital for selecting agronomic traits, molecular markers provide complementary insights into genetic potential. However, few studies integrate both approaches for Egyptian maize genotypes, despite the country reliance on newly reclaimed arid lands for production. This gap is critical, as region-specific challenges, such as low rainfall and sandy soils, demand locally adapted genotypes. To address this, the present study employed SCoT, CDDP, and SRAP markers, chosen for their distinct advantages: SCoT targets functional gene regions linked to stress responses, CDDP detects polymorphisms in conserved drought-related genes, and SRAP captures coding-region diversity associated with yield traits. By combining field evaluations of 14 inbred lines across three seasons with this multi-marker system, this study connects phenotypic performance and genetic divergence, offering a dual-strategy framework to accelerate breeding for arid zones. Therefore, this work primarily aimed to assess the variability and stability of critical agronomic traits focusing on phenological traits, plant stature, ear characteristics, and grain yield among diverse maize inbred lines, explore the relationships between yield and its attributes, and classify the inbred lines based on their agronomic performance to guide utilization in hybrid development. In addition to exploring genetic diversity among the inbred lines employing three molecular marker

techniques: SCoT, CDDP, and SRAP. Utilizing a combined approach of agronomic and molecular evaluation provides significant insights into the genetic potential for future breeding programs.

## MATERIALS AND METHODS

### Plant materials and field trial

Fourteen newly developed maize inbred lines (*Zea mays* L.) were obtained from the Agricultural Research Center, ARC, Egypt for this study. The names, origin and pedigree of these lines are presented in Table S1. Based on previous screening, the inbred lines were selected to represent contrasting phenological and morphological characteristics, such as differences in maturity, plant height, and ear characteristics. In addition, priority was given to lines that exhibited resistance to prevalent local pests and diseases, including pink stem borer (*Sesamia cretica*), purple-lined borer (*Chilo agamemnon*), and late wilt disease. The assessed inbred lines were evaluated in a three-year field trial conducted during the summer seasons of 2021, 2022, and 2023 at Mit Yazid, Menia Al-Qamh, Sharqia Governorate, Egypt (30°30′36″N, 31°19′48″E). The experimental site features an arid, hot climate with no precipitation during the summer months. The experimental site experienced high temperatures with monthly minimum temperatures ranged from 17.96 °C to 26.12 °C and maximum temperatures from 32.83 °C to 39.45 °C. Growing degree days varied between 378 and 599 °C per month. Relative humidity remained moderate, at 63–70%. Precipitation was extremely low, with most months recording 0–1.36 mm. These conditions were warmer and drier than the 40-year average, reflecting a typical arid climate. Table S2 presents the monthly average temperatures, growing degree days, relative humidity, and total rainfall of the three seasons alongside the long-term averages over the last four decades. The experimental site soil has a clay texture (49.12% clay, 29.26% silt, and 21.62% sand) with moderate alkaline properties (pH = 8.01). A randomized complete block design was used, with three replicates per season. Sowing was conducted during the second week of May in all three seasons, aligning with the region's recommended planting time for maize. The plots contained three rows, each five m in length, with a row spacing of 70 cm and a plant spacing of 25 cm. Potassium, phosphorus, and nitrogen fertilization were added at 116 kg/$K_2O$/ha, 76 kg $P_2O_5$/ ha, and 290 N/ha. Data on days to 50% tasseling and silking were documented for each plot. At harvest, ten plants were randomly chosen from each plot to measure plant height, ear height, ear length, ear diameter, number of rows per ear, and number of kernels per row. The plots were harvested manually, and grain yield per plant was calculated by adjusting the weight of the shelled grain to a moisture content of 15.5%. Furrow irrigation was applied following the practice of the region of the study. Throughout each growth season, the irrigation provided about 6,500–7,500 m$^3$/ha at intervals of 10–12 days.

### Molecular analyses
#### DNA extraction

Genomic DNA was extracted from young leaves of 14 maize inbred lines using a modified CTAB extraction protocol (*Doyle, 1991*). DNA quality and quantity were assessed using a

**Table 1  SCoT, SRAP and CDDP primers with their nucleotide sequences, molecular weight g/mol, primer length, and GC% content.**

| Primers | Sequences (5′–3′) | Molecular weight g/mol | Primer length | GC% content |
|---|---|---|---|---|
| SCoT1 | ACCATGGCTACCACCGGC | 5,429.6 | 18 | 66.67% |
| SCoT2 | CAATGGCTACCACTACAG | 5,452.6 | 18 | 50.00% |
| SCoT3 | ACAATGGCTACCACTGCC | 5,428.6 | 18 | 55.56% |
| SCoT4 | CAACAATGGCTACCACCG | 5,437.6 | 18 | 55.56% |
| SCoT5 | ACCATGGCTACCACGGCA | 5,453.6 | 18 | 61.11% |
| SCoT6 | CAACAATGGCTACCACCC | 5,397.6 | 18 | 55.56% |
| SCoT7 | ACCATGGCTACCACCGAG | 5,453.6 | 18 | 61.11% |
| SCoT8 | CACCATGGCTACCACCAG | 5,413.6 | 18 | 61.11% |
| SCoT9 | ACCATGGCTACCACCGTG | 5,444.6 | 18 | 61.11% |
| CDDP1 | ATGGGCCGSGGCAAGGTGG | 5,645.7 | 19 | 72.22% |
| CDDP2 | GGCAAGGGCTGCCGC | 4,619.0 | 15 | 80.00% |
| SRAP1 | F: TGAGTCCAAACCGGAAG | 5,228.5 | 17 | 52.94% |
|  | R: GACTGCGTACGAATTTGC | 5,514.7 | 18 | 50.00% |
| SRAP2 | F: TGAGTCCAAACCGGATA | 5,203.5 | 17 | 47.06% |
|  | R: GACTGCGTACGAATTAAC | 5,507.7 | 18 | 44.44% |

NanoDrop 2000 spectrophotometer (Thermo Fisher Scientific, Waltham, MA, USA). The integrity of the DNA was verified by electrophoresis on 1% agarose gel.

### Start codon targeted

A 25 μL SCoT-PCR reaction was performed using GoTaq Green Master Mix, 100 ng template DNA, and specific primers. The PCR cycling conditions were as follows: initial denaturation at 94 °C for 5 min, followed by 40 cycles of 94 °C for 30 s, 72 °C for 1 min, and a final extension at 72 °C for 10 min. PCR amplification was performed using a Bio-Rad T100 Thermal Cycler.

### Conserved DNA-derived polymorphism

A 25 μL PCR reaction was conducted using 10 μL GoTaq Green Master Mix, 1 μL each of the forward and reverse SRAP primers (Table 1), two μL of 100 ng genomic DNA template, and 25 μL nuclease-free water. Amplification was performed using a Bio-Rad T100 Thermal Cycler using the following conditions: initial denaturation at 94 °C for 5 min, followed by 40 cycles of denaturation at 94 °C for 30 s, annealing at 72 °C for 1 min, and a final extension at 72 °C for 10 min.

### Sequence-related amplified polymorphism

A 25 μL SRAP-PCR reaction was performed employing 10 μL GoTaq Green Master Mix, one μL of each forward and reverse primer (listed in Table 1), two μL of 100 ng genomic DNA, and 25 μL nuclease-free water. PCR cycling conditions were as follows: an initial denaturation step at 94 °C for 5 min, followed by 40 cycles of denaturation at 94 °C for 30 s, annealing at 72 °C for 1 min, and a final extension at 72 °C for 10 min, using a Bio-Rad T100 Thermal Cycler (Bio-Rad, Hercules, CA, USA).

### Gel electrophoresis

PCR products (10 μl) were electrophoresed on a 2% agarose gel stained with 0.5 μg/ml ethidium bromide and visualized using an Aplegen Omega Lum G gel documentation system (Aplegen, Pleasanton, CA, USA). A 100 bp plus DNA ladder (Fermentas, Waltham, MA, USA) was used as a molecular weight marker.

## Data analysis

A least significant difference test was performed at $p \leq 0.05$. In addition, principal component analysis, cluster analysis, and heatmap were performed using R software (version 4.2.2). The R packages FactoMineR and factoextra were utilized for principal component analysis, pheatmap and gplots for heatmap clustering. For molecular data analysis, SCoT, CDDP, and SRAP marker profiles were scored as 1 for the presence and 0 for the absence of individual alleles. The polymorphism level, a measure of genetic variation, was determined by dividing the number of loci exhibiting polymorphism (different banding patterns) by the total number of scored loci. Genetic similarities among the maize cultivars and hybrids were computed using Dice's coefficient (*Dice, 1945*). This coefficient was determined utilizing SPSS software version 29.0.10 (IBM Corp., Armonk, NY, USA; *Norušis, 1993*). A clustering analysis was subsequently performed to generate a dendrogram depicting the phylogenetic relationships among the inbred lines (*Rokach & Maimon, 2005*).

## RESULTS

### Agronomic performance
### Phenological traits

The evaluated inbred lines displayed significant differences across phenological traits over the three growing seasons. In the first season, days to tasseling ranged from 47.0 to 74.3, averaging 69.6 days. During the second season, values ranged from 48.7 to 69.7 days, with an average of 67.0 days, while in the third season, the range was 46.7 to 67.0 days, averaging 62.9 days. IKA22 consistently exhibited the earliest tasseling across all three growing seasons (Fig. 1A). In the first season, IKA22 was followed by DKCA2, LZAM7B, B17AB, LA442B, and LMP214A, while in the second and third seasons, LA442B, ZBM40A, LMP214A, and RA28C exhibited the next earliest tasseling (Fig. 1A).

Days to silking fluctuated from 49.0 to 76.7 in the first season, averaging 72.0 days. In the second season, values ranged from 51.0 to 71.7 days, with an average of 68.8 days, while in the third season, the range was 49.3 to 68.7 days, averaging 65.0 days. The earliest silking line was consistently IKA22, followed by DKCA2, LZAM7B, B17AB, and LA442B in the first season and LA442B, ZBM40A, LMP214A, and RA28C in the second and third seasons (Fig. 1B). Lines with the longest silking periods included LZP210, DKC14, SNY23, SSK36, and LCM54 in the first season, LZP210, DKCA2, DKC14, LZAM7B, and B17AB in the second, and LCM54, SNY23, B17AB, LZP210, and SSK36 in the third (Fig. 1B).

### Plant stature

Plant height varied significantly across the seasons. In the first season, values fluctuated from 170.0 to 246.7 cm, averaging 195.3 cm. During the second season, the range narrowed

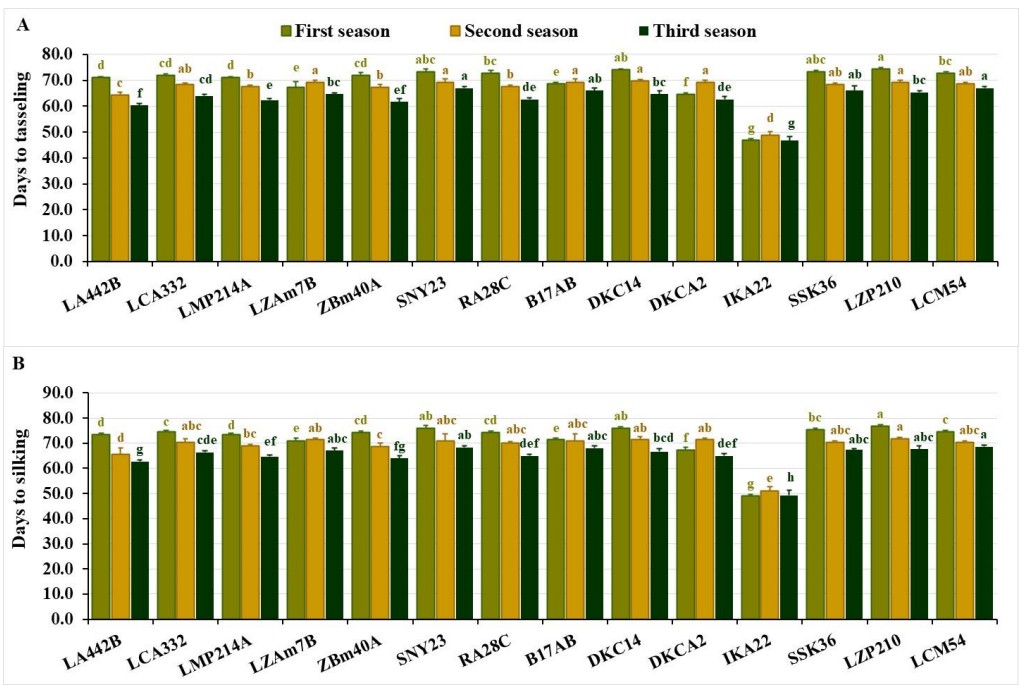

**Figure 1** **Phenological characteristics of the assessed inbred lines.** (A) Days to tasseling and (B) days to silking. The bars above the columns indicate standard deviations (SD). Columns in the same color with distinct letters reveal substantial differences as determined by LSD with a *p*-value of less than 0.05.

to 150.0–200.0 cm, averaging 179.0 cm. In the third season, heights spanned from 143.3 to 221.7 cm, averaging 175.0 cm. The shortest lines across the seasons were IKA22, LZP210, B17AB and ZBM40A in the first season, B17AB, SNY23, IKA22 and SSK36 in the second, and LZP210, IKA22, B17AB, LZAM7B, and SSK36 in the third (Fig. 2A). Conversely, the tallest lines included SNY23, LCM54, RA28C, LCA332, DKC14 in the first season, LZP210, LCM54, ZBM40A, DKC14, and RA28C, in the second, and RA28C, DKC14, SNY23, LCA332, and LCM54 in the third (Fig. 2A).

Ear height also showed significant variation across seasons. In the first season, values ranged from 75.0 to 121.7 cm, averaging 94.5 cm. During the second season, the range was 65.0–95.0 cm, with an average of 79.8 cm. In the third season, ear height spanned from 56.7 to 88.3 cm, averaging 71.0 cm. Lines with the lowest ear positions included LA442B, LZP210, DKC14, IKA22, and SSK36 in the first season, IKA22, SSK36, SNY23, B17AB, and LZP210 in the second, and SSK36, LZAM7B, B17AB, IKA22, and LMP214A in the third (Fig. 2B). In contrast, lines with the highest ear positions were LCM54, SNY23, RA28C and LMP214A in the first season, ZBM40A, LCM54, DKC14, RA28C, and LA442B in the second, and RA28C, SNY23, DKC14, ZBM40A, and LCM54 in the third (Fig. 2B).

### Ear characters

Ear diameter differed from 3.15 to 3.88 cm in the first season, averaging 3.50 cm. During the second and third seasons, values were consistent, spanning 3.07–3.80 cm and
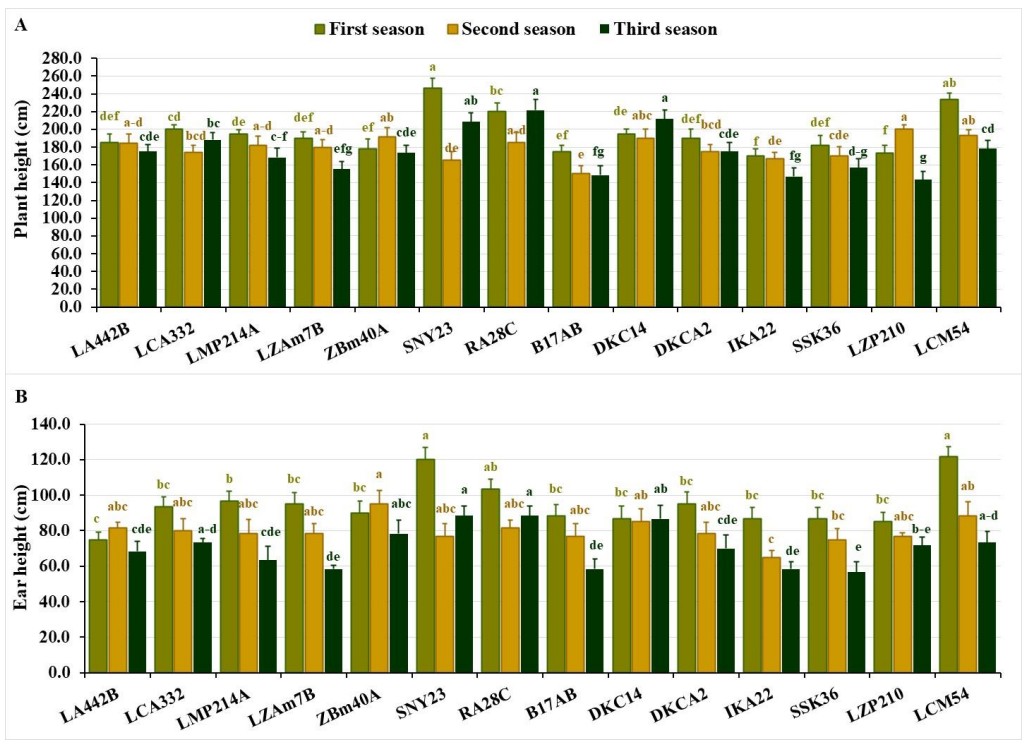

**Figure 2 Plant stature characteristics of the assessed inbred lines.** (A) Plant height and (B) ear height (cm). The bars above the columns indicate standard deviations (SD). Columns in the same color with distinct letters reveal substantial differences as determined by LSD with a *p*-value of less than 0.05.

3.17–3.73 cm, respectively, with averages of 3.50 cm. In the first season, the lines LCA332, RA28C, ZBM40A, LZP210, and DKC14 exhibited the largest diameters. While, in the second season, IKA22, RA28C, LCA332, ZBM40A, and LA442B performed best, while in the third season, ZBM40A, IKA22, RA28C, B17AB, and LZP210 recorded the highest values (Fig. 3A). Similarly, the lines with the smallest ear diameters varied across the seasons. In the first season, LCM54, IKA22, SSK36, SNY23, and B17AB revealed the lowest diameter values, while in the second season, the lines SNY23, SSK36, LZP210, B17AB, and DKC14 exhibited the smallest diameters. During the third season, the lowest values were observed in LCM54, LA442B, LZAM7B, LMP214A, and SNY23 (Fig. 3A).

Ear length showed significant variation across the seasons, ranging from 12.42 to 17.38 cm in the first season (average 15.60 cm), 12.43 to 17.80 cm in the second (average 15.20 cm), and 13.10 to 16.20 cm in the third (average 14.70 cm). Top-performing lines included LCA332, RA28C, SNY23, ZBM40A, and LZAM7B in the first season, RA28C, LCM54, SNY23, LZAM7B, and LMP214A in the second, and B17AB, ZBM40A, RA28C, DKCA2, and SSK36 in the third (Fig. 3B). The lines with the shortest ear lengths also showed seasonal variation. In the first season, the shortest ear lengths were observed in LA442B, IKA22, B17AB, SSK36, and LMP214A. Similarly, in the second season, the lines B17AB, IKA22, LA442B, SSK36, and DKCA2 exhibited the smallest ear lengths, while in
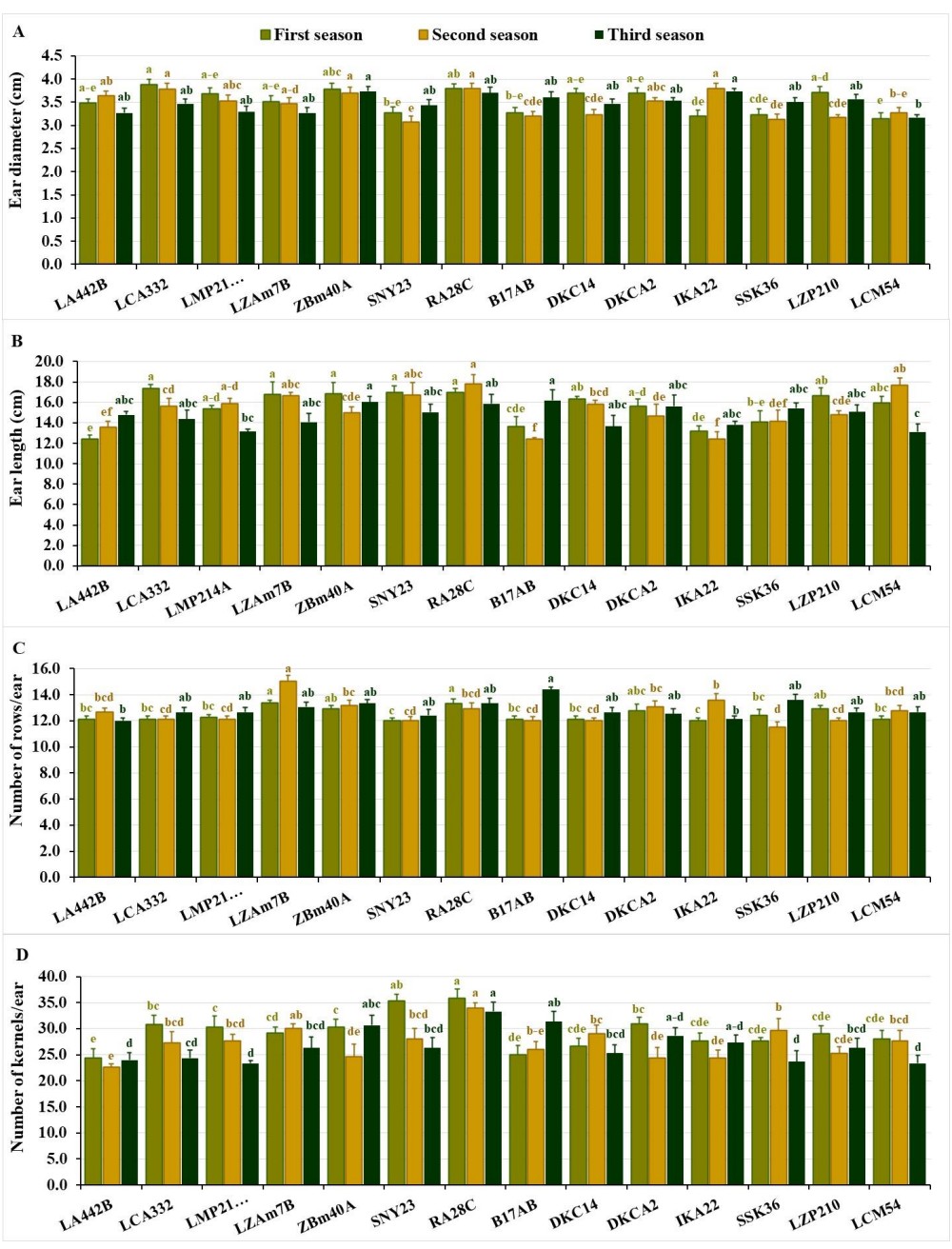

**Figure 3  Ear characteristics of the assessed inbred lines.** (A) Ear diameter, (B) ear length, (C) number of rows/ear; and (D) number of kernels/ear. The bars above the columns indicate standard deviations (SD). Columns in the same color with distinct letters reveal substantial differences as determined by LSD with a *p*-value of less than 0.05.

the third season, the shortest values were recorded for LCM54, LMP214A, DKC14, IKA22, and LZAM7B (Fig. 3B). The number of rows/ear displayed significant differences across the seasons. In the first season, values fluctuated from 12.00 to 13.37, averaging 12.50

rows per ear. In the second season, the number of rows/ear varied between 11.53 and 15.07, averaging 12.70 rows, while in the third season, values altered from 12.00 to 14.40, with an average of 12.90 rows/ear. Lines with the greatest number of rows during the first season were LZAM7B, RA28C, LZP210, ZBM40A, and DKCA2. In the second season, the top-performing lines included LZAM7B, IKA22, ZBM40A, DKCA2, and RA28C, while in the third season, B17AB, SSK36, RA28C, ZBM40A, and LZAM7B demonstrated the highest number of rows/ear (Fig. 3C). In contrast, in the first season, the lines with the fewest rows/ear were SNY23, IKA22, LA442B, LCA332, and B17AB. Similarly, SSK36, SNY23, B17AB, DKC14, and LZP210 recorded the lowest values in the second season, while LA442B, IKA22, SNY23, DKCA2, and LCA332 exhibited the lowest numbers in the third season (Fig. 3C).

The number of kernels/row varied significantly across the three seasons. In the first season, values changed from 24.33 to 35.87 kernels per row, averaging 29.40 kernels. During the second season, the range was 22.67 to 34.00 kernels, with an average of 27.20 kernels, while in the third season, values differed from 23.33 to 33.33 kernels, with an average of 26.70 kernels. The highest-performing lines in terms of kernels per row during the first season were RA28C, SNY23, DKCA2, LCA332, and ZBM40A, while in the second season, the top lines included RA28C, LZAM7B, SSK36, DKC14, and SNY23. In the third season, the lines RA28C, B17AB, ZBM40A, DKCA2, and IKA22 exhibited the greatest number of kernels per row (Fig. 3D). During the first season, lines with the fewest kernels per row were observed in LA442B, B17AB, DKC14, IKA22, and SSK36. Similarly, LA442B, DKCA2, IKA22, ZBM40A, and LZP210 recorded the lowest values in the second season, while LMP214A, LCM54, SSK36, LA442B, and LCA332 exhibited the lowest numbers in the third season (Fig. 3D).

### Grain yield

Grain yield per plant exhibited substantial variation across the three seasons. In the first season, yield fluctuated from 47.07 to 118.67 g, with an average of 79.8 g. In the second season, values ranged from 56.93 to 87.43 g (average 72.4 g), while in the third season, yield spanned 75.33 to 131.80 g, averaging 105.4 g. Lines with superior yields included DKCA2, RA28C, ZBM40A, LZAM7B, and LCA332 in the first season, LA442B, RA28C, SSK36, ZBM40A, and LZAM7B in the second, and RA28C, B17AB, ZBM40A, IKA22, and DKC14 in the third (Fig. 4). Conversely, lines with the lowest grain yield varied across seasons. In the first season, the lines IKA22, SSK36, B17AB, DKC14, and LA442B recorded the lowest yields. In the second season, SNY23, B17AB, LZP210, LCM54, and LMP214A produced the lowest yields, while LCM54, LCA332, LMP214A, LA442B, and SNY23 produced the lowest yields in the third season (Fig. 4).

## Grouping of assessed inbred lines based on yield characters

The evaluated inbred lines were clustered into six distinct groups based on their yield-related traits, as shown in Fig. 5. Group A comprised a single line, RA28C, which exhibited superior performance in grain yield and its contributing traits. Group B included three lines (LZAM7B, ZBM40A, and DKCA2) characterized by intermediate-to-high yield traits.

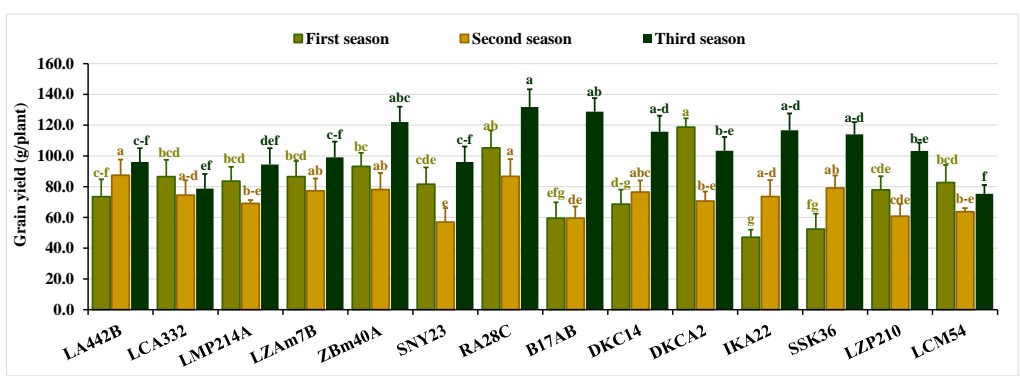

**Figure 4** **Grain yield of the assessed inbred lines.** The bars above the columns indicate standard deviations (SD). Columns in the same color with distinct letters reveal substantial differences as determined by LSD with a *p*-value of less than 0.05.

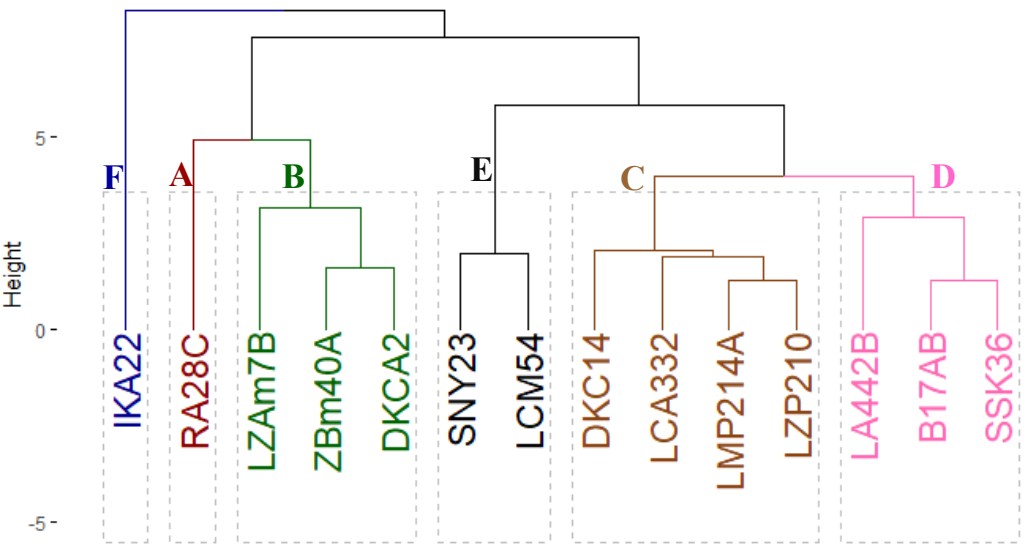

**Figure 5** **Cluster displays the phenotypic distances among the fourteen inbred lines based on their agronomic performance.**

Group C comprised four lines (DKC14, LCA332, LMP214A, and LZP210) that displayed intermediate-to-low yield traits. Group D contained three lines (LA442B, B17AB, and SSK36) with consistently low yield traits. Lastly, Groups E and F included two lines (SNY23 and LCM54) and one line (IKA22), all demonstrating the lowest yield traits.

## Association among lines and agronomic traits

Principal component analysis (PCA) was utilized to explore the associations between maize lines and the evaluated agronomic traits. The first two principal components (PCs)

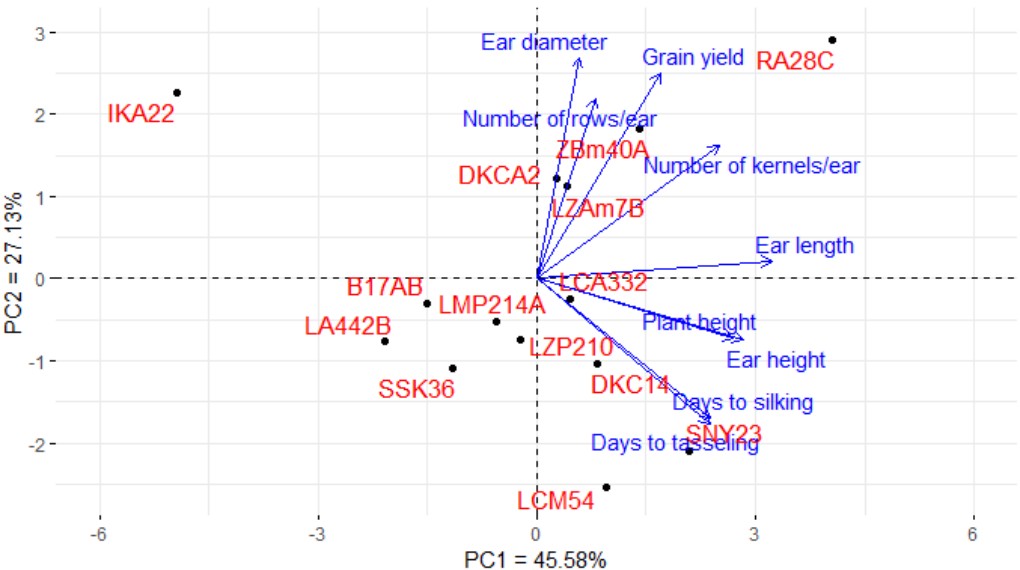

**Figure 6 PC-biplot explores the association between maize inbred lines and agronomic traits.**

elucidated a considerable proportion of total variance, with PC1 accounting for 45.58% and PC2 for 27.13%, as illustrated in the PCA biplot (Fig. 6). PC1 captured the most significant variation, effectively distinguishing hybrids with positive and negative values. Certain lines were positioned on the positive side of PC1 and related to yield traits, such as RA28C, ZBM40A, LZAM7B, and DKCA2. Conversely, lines on the negative side, such as IKA22, exhibited inferior yield traits. The closeness of trait vectors in the PCA biplot indicates a robust correlation between grain yield and several key characters, notably the number of kernels per row, number of rows per ear, and ear diameter. Figure 7 shows the heatmap of the evaluated agronomic traits, revealing distinct clustering patterns among the maize inbred lines. In the heatmap, blue indicates the highest trait values, while red denotes the lowest values, enabling visual differentiation of lines with superior and inferior agronomic performance. This clustering highlights the genetic and phenotypic diversity present among the studied lines. Lines such as RA28C, ZBM40A, DKCA2, and LZAM7B displayed the highest values (represented in blue), whereas IKA22 displayed the lowest trait values (highlighted in red).

## Molecular analyses

Genetic diversity among 14 maize inbred lines was assessed using three molecular marker techniques: SCoT (Fig. 8), CDDP, and SRAP (Fig. 9). A total of 95 loci were amplified using SCoT, CDDP, and SRAP primers (Table 2). An average of 7.3 loci were amplified per primer. Among the 95 SCoT, CDDP, and SRAP loci, 74 were polymorphic (5.69/primer), and 21 were monomorphic (1.61/primer). The percentage of polymorphic loci ranged from 0% (SRAP1) to 100% (SCoT2, 3, and 9), averaging 72.32%. Genetic distances, calculated using the Dice coefficient, ranged from 4.0 to 6.48. The lowest distance was observed between LZAM7B and ZBM40A, as well as between LZP210 and LCM54 (Table S3).

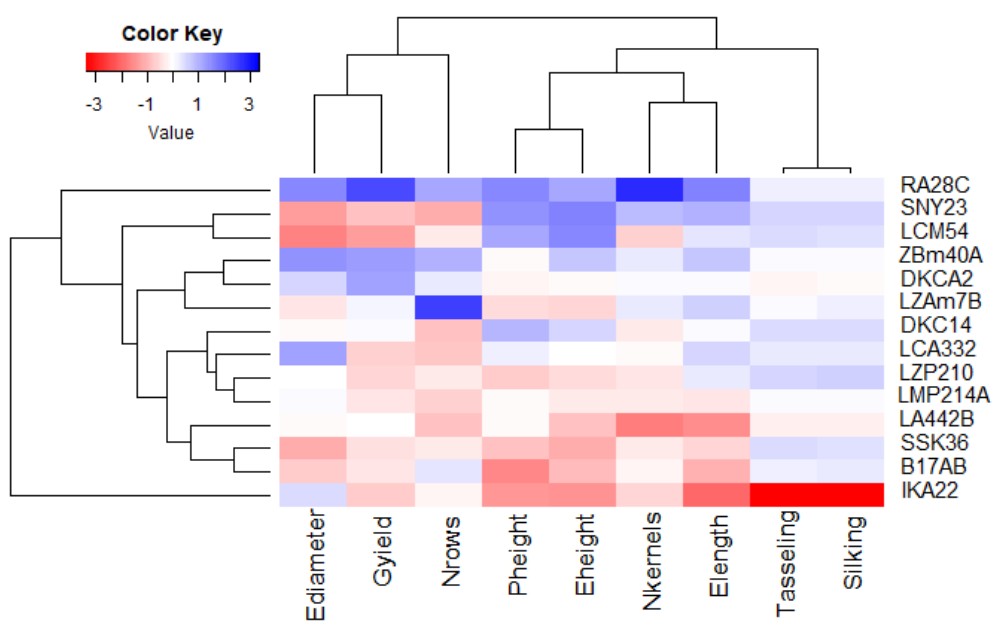

**Figure 7** Heatmap analysis revealing the relationships among agronomic attributes in assessed inbred lines.

Conversely, the highest distance was found between LZAM7B and DKCA2. According to Table S4, the highest similarity (0.86) was observed between LZAM7B and ZBM40A, whereas the lowest similarity (0.60) was found between LZAM7B and DKCA2.

The clustering analysis based on SCoT, CDDP and SRAP banding profiles grouped the evaluated maize inbred lines into five groups (Fig. 10). Cluster A consisted solely of DKCA2, while Cluster B included IKA22, SSK36, LZP210, and LCM54. Cluster C was represented exclusively by LA442B, and Cluster D encompassed four genotypes: LMP214A, DKC14, RA28C, and B17AB. Lastly, Cluster E comprised four genotypes LCA332, LMP214A, LZAM7B, ZBM40A, and SNY23.

# DISCUSSION

Evaluating maize inbred lines across multiple growing seasons provides a comprehensive understanding of their agronomic performance, genetic variability and stability. Furthermore, incorporating molecular marker analysis to assess genetic diversity enhances the ability to identify potential parents in hybrid development. This combined approach of agronomic and molecular evaluation provides valuable insights into the genetic potential and adaptability of maize genotypes. Such evaluations are essential for identifying superior lines with desirable traits to contribute significantly to maize breeding programs. This integrated strategy is crucial for improving maize productivity, resilience, and genetic gains in future breeding efforts, particularly under the current challenges posed by climate fluctuations.

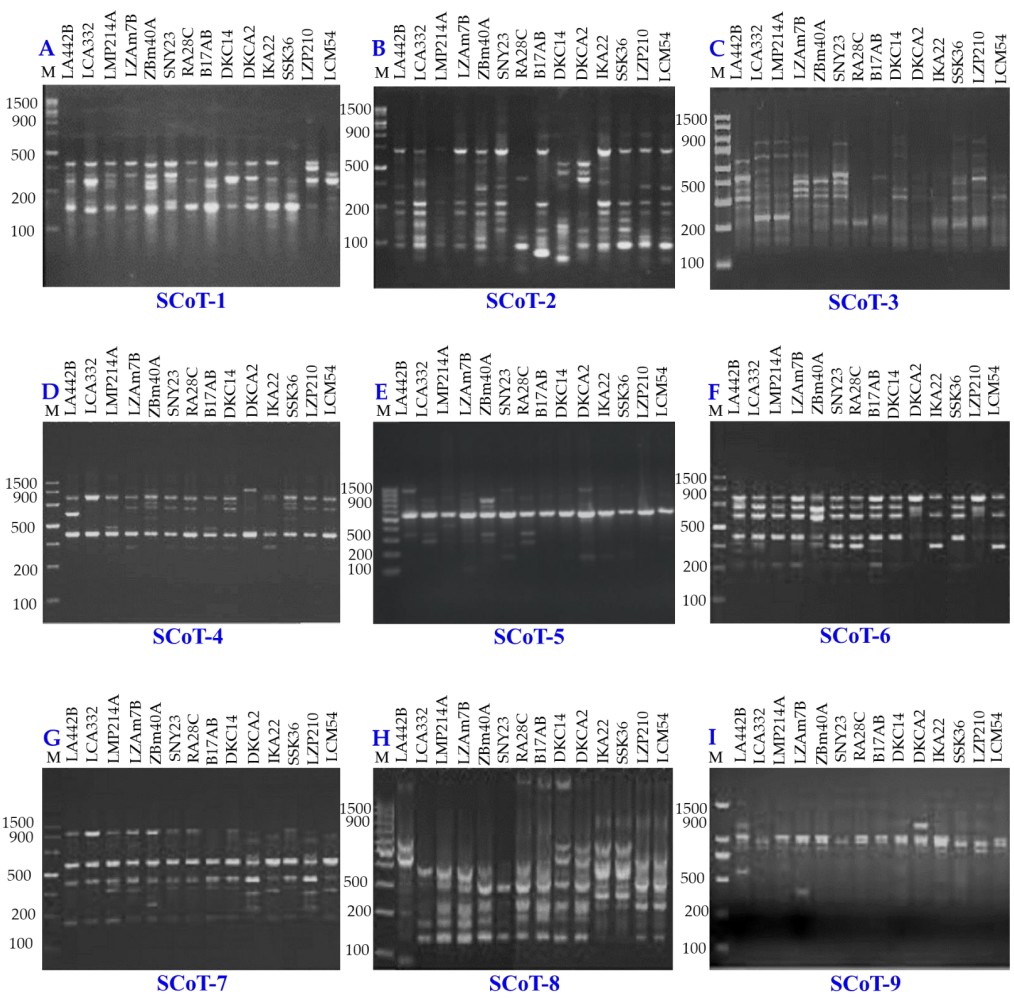

**Figure 8** **SCoT-PCR amplification patterns of 14 maize inbred lines using nine SCoT primers (A–I).** M = 100 bp plus DNA ladder.

The field evaluation of maize lines over three growing seasons revealed significant variability across phenological attributes, plant stature, ear characteristics, and grain yield. These findings explored the diversity in the assessed germplasm and its implications for breeding programs to enhance maize productivity. The significant differences observed in phenological traits, including days to tasseling and silking across seasons, emphasized the diverse genetic makeup and adaptability of the inbred lines. Early maturing line IKA22 is valuable for breeding programs targeting environments with shorter growing seasons or limited water availability. Conversely, late-maturing lines like LZP210 and DKC14 may be better suited for environments with extended growing periods. The season-to-season fluctuation in phenological traits also reflects the impact of environmental features on the expression of these traits, revealing the importance of multi-season evaluations in identifying stable and adaptable genotypes. In this perspective, *Shao et al. (2021)*, *Zhou et al. (2021)* and *Wang et al. (2020)* manifested that days to tasseling and silking are impacted

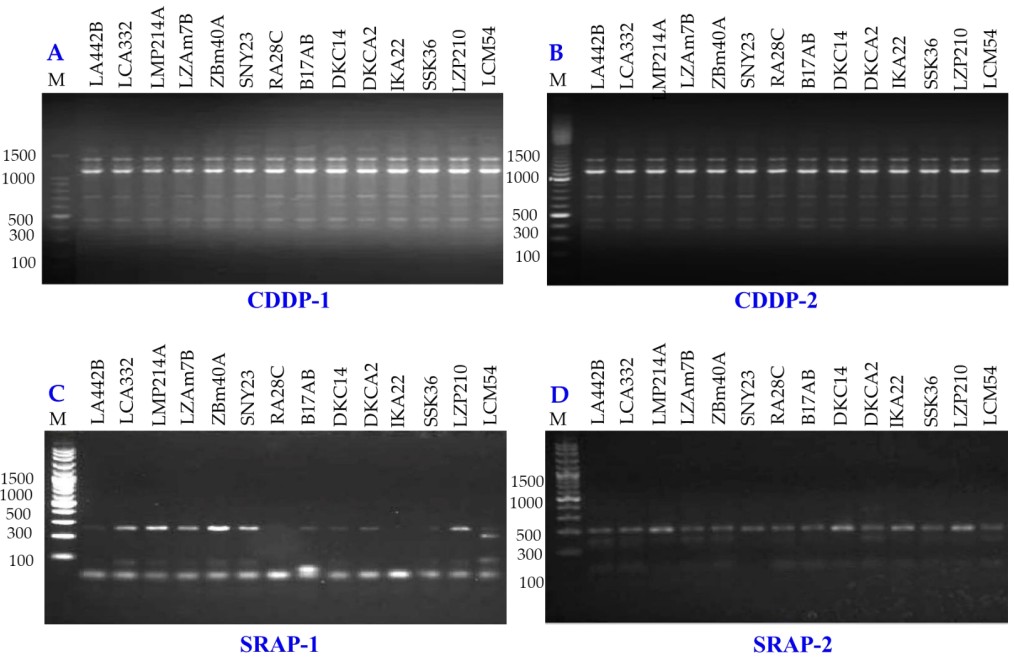

**Figure 9  CDDP and SRAP-PCR amplification patterns of 14 maize inbred lines using CDDP (A, B) and SRAP primers (C, D).** M = 100 bp plus DNA ladder.

**Table 2  Primers, number of monomorphic bands, number of polymorphic bands, percentage of polymorphism generated by SCoT, CDDP and SRAP-PCR amplification patterns in 14 Egyptian maize genotypes.**

| Primer | Total number of loci | Number of monomorphic loci | Number of polymorphic loci | Number of unique loci | Polymorphism (%) |
|---|---|---|---|---|---|
| SCoT1 | 8 | 1 | 7 | 1 | 88% |
| SCoT2 | 14 | 0 | 14 | 0 | 100% |
| SCoT3 | 9 | 0 | 9 | 0 | 100% |
| SCoT4 | 8 | 1 | 7 | 2 | 88% |
| SCoT5 | 8 | 1 | 7 | 3 | 88% |
| SCoT6 | 6 | 1 | 5 | 0 | 83% |
| SCoT7 | 8 | 3 | 5 | 0 | 63% |
| SCoT8 | 8 | 1 | 7 | 0 | 88% |
| SCoT9 | 7 | 0 | 7 | 3 | 100% |
| CDDP1 | 7 | 6 | 1 | 0 | 14% |
| CDDP2 | 5 | 1 | 4 | 2 | 80% |
| SRAP1 | 5 | 5 | 0 | 0 | 0% |
| SRAP2 | 2 | 1 | 1 | 0 | 50% |
| Total | 95 | 21 | 74 | 11 | |
| Average | 7.3 | 1.61 | 5.69 | 0.84 | 72.46% |

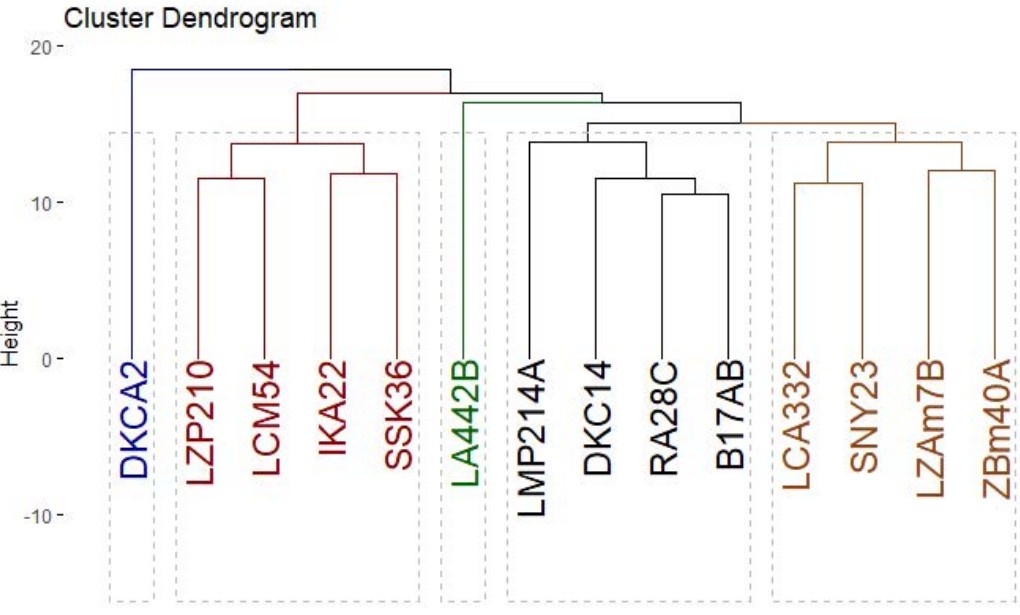

**Figure 10** Phylogenetic tree of 14 Egyptian maize inbred lines was revealed based specific SCoT, CDDP and SRAP-based markers data.

by environmental factors, such as light, temperature, and soil fertility, alongside genetic factors. The variability in these traits across different growing seasons, as observed in the present study, aligns with previous findings from research on maize phenology. *Shao et al. (2021)* and *Lv et al. (2024)* elucidated that days to tasseling and silking in maize are primarily governed by genetic makeup and environmental conditions, with early maturing lines exhibiting shorter tasseling and silking. Additionally, *Comas et al. (2019)* emphasized the importance of early maturing lines in optimizing maize productivity, especially in areas prone to drought or where the growing season is limited. Furthermore, identified lines with consistent early maturity also correspond to *Mugiyo et al. (2021)* who found that early maturing hybrids are crucial for ensuring timely harvests and minimizing the risk of crop failure due to unfavorable late-season weather.

Ear height and plant height are essential agronomic characteristics closely linked to the yield potential in maize. The considerable variation observed in these traits reflects the genetic diversity among the evaluated lines in plant stature. Shorter lines, such as IKA22, B17AB, and SSK36 may be advantageous in breeding programs aiming to develop lodging-resistant hybrids. In contrast, taller lines like RA28C and LCM54 could be utilized for traits associated with more efficient photosynthesis, high biomass, and increased yield potential. The consistency of certain lines across seasons for these traits indicates genetic stability, which is critical for developing resilient maize hybrids. *Zhu et al. (2023)* depicted that tall maize hybrids exhibit better yields due to their greater leaf area and improved photosynthetic efficiency. Conversely, *Rehman et al. (2024)* manifested that excessively tall plants can lead to poor root anchorage and increased lodging, which may offset the benefits

of taller plants. However, the current study demonstrated that the tall lines also exhibited strong performance in terms of yield, suggesting that these lines could balance height with yield potential.

The variation in ear diameter, ear length, and number of rows and kernels per ear highlights the unique roles these traits play in overall yield. Larger ear diameters and longer ears are often associated with higher grain production, and the number of rows/ear directly affects the total number of kernels produced. The inbred lines RA28C, ZBM40A, DKCA2, and LZAM7B consistently exhibit superior ear characteristics, suggesting their potential as parents in hybrid development. Conversely, lines with inferior ear traits, such as IKA22 and LCM54, may require improvement through the introgression of favorable alleles. The strong associations detected by PCA between these traits and grain yield further reinforced their importance as selection criteria in breeding programs. The relationship between these traits and grain yield has been documented in previous reports. *Kara (2011)* and *Rotili et al. (2022)* demonstrated that increased ear diameter is directly related to kernel number and subsequently grain yield. Similarly, *Kandil, Sharief & Abozied (2017)* disclosed that a higher number of rows per ear leads to a considerable enhancement in grain yield, particularly in environments with optimal growing conditions.

Grain yield is the primary factor determining maize performance, and both genetic and environmental factors often influence its variability across seasons. In this study, grain yield exhibited substantial seasonal variations, reflecting the differential response of assessed inbred lines to ecological conditions. In this perspective, *Kim & Lee (2023)* and *Rajath et al. (2023)* elucidated that maize yield is highly vulnerable to environmental conditions such as soil fertility, water availability, and temperature fluctuations. Detected high-yielding lines like RA28C, ZBM40A, DKCA2, and LZAM7B consistently performed well in multiple seasons, indicating their suitability for high-yielding hybrid combinations. This is consistent with *Nhantumbo et al. (2021)* and who suggested that evaluating yield stability in multi-season trials is essential for developing genotypes performing well under varying conditions. Otherwise, low-yielding lines, such as IKA22 and B17AB, may be less adaptable but could still have unique traits beneficial for specific breeding objectives to enhance tolerance for biotic and abiotic stresses (*Di Matteo et al., 2016*; *Masuka et al., 2017*).

Cluster and PCA analyses have become essential tools for understanding the relationships between agronomic traits and grouping lines based on their performance. Clustering inbred lines based on yield-related characteristics could provide a practical framework for maize breeding. Classification of lines into six distinct clusters based on yield-related traits offered valuable insights into the genetic potential of each line for further breeding programs. Separating high-yielding lines into different clusters, such as Group A (RA28C) and Group B (LZAM7B, ZBM40A, and DKCA2), revealed their superior performance and potential for hybrid development. Conversely, the low-yielding lines such as IKA22 (Group F), showed the need for targeted genetic improvement to enhance their yield potential. Similarly, *Omar et al. (2022)*, and *Yue et al. (2022a)* utilized cluster analysis to group maize hybrids based on yield and associated traits, finding that genotypes with similar agronomic

characteristics could be grouped into distinct clusters that exhibited similar performance across multiple environments.

The seasonal variation observed in the agronomic traits of the maize inbred lines provided important insights into the potential implications for breeding, particularly in arid conditions. The results revealed significant fluctuations in the studied agronomic traits across the three growing seasons, emphasizing the importance of understanding how environmental factors influence maize performance. The observed differences in days to tasseling and silking across seasons suggest that certain inbred lines exhibit more consistent early flowering, which may be advantageous in regions where early maturation is crucial for overcoming drought stress. Moreover, the variation in plant stature and ear characteristics indicated the need for breeders to focus on traits that contribute to tolerance to environmental stresses. The seasonal fluctuations in grain yield revealed the necessity of selecting lines that can adapt to varying climatic conditions while maintaining high productivity. These findings align with previously published studies indicating that maize performance is highly sensitive to environmental factors (*Hama & Mohammed, 2019*; *Yue et al., 2022b*; *Yue et al., 2025*).

The agronomic traits, including phenological traits, plant stature, and ear characteristics, provide crucial insights into the practical considerations for breeding programs aimed at improving maize resilience, particularly in arid areas. The variation in phenological traits, such as days to tasseling and silking, has direct implications for breeding early maturing maize hybrids, which are critical for ensuring that crops can reach maturity before water stress intensifies. Lines like IKA22, which consistently exhibited early tasseling and silking, could be exploited in breeding programs to develop hybrids with shorter growing periods, enhancing their adaptability to regions with unpredictable rainfall or shorter growing seasons. In addition to phenological traits, plant stature and ear height revealed the importance of optimizing plant architecture for better resource use efficiency. Shorter plants, like IKA22 and LZP210, tend to use water more efficiently and may be less prone to lodging, making them valuable candidates for breeding programs targeting water-limited environments. Similarly, understanding the variation in ear characteristics such as ear diameter, length, and the number of kernels per row allows breeders to select for lines that produce higher-quality grains. For instance, lines like RA28C and ZBM40A, which consistently performed well in terms of ear diameter and kernel number, could be selected for breeding to enhance yield potential even in suboptimal conditions. Overall, these agronomic traits could be considered important indicators for improved yield under environmental stresses and could inform breeding strategies to meet the challenges of climate change. By linking specific traits to their practical significance, breeders can more effectively develop maize hybrids that are both high-yielding and resilient to environmental stress ensuring food security.

PCA, as a powerful tool for visualizing complex data, was used to analyze the association between assessed lines and evaluated agronomic traits (*ElShamey et al., 2022*; *Mansour et al., 2021*; *Salem et al., 2020*). The PCA results supported that RA28C, ZBM40A, DKCA2, and LZAM7B were positioned on the positive side of the first principal component, indicating their strong association with higher yield traits. Additionally, the heatmap

reinforced the findings of the PCA, providing further evidence of the genetic divergence among the assessed lines. Moreover, the PCA results provided a comprehensive view of trait interrelationships and their impact on yield. The strong positive associations between grain yield and characteristics such as ear diameter, number of kernels per row, and number of rows per ear suggest that these parameters are critical determinants of productivity. The distinct separation of high- and low-performing lines in the PCA biplot further supports their effectiveness as key selection criteria in maize breeding. *Zhang et al. (2023)*, *Sedhom et al. (2024)*, *Ghazy et al. (2024)* and *Galal et al. (2025)* used PCA to determine the correlation between several traits and their contribution to overall yield in maize, finding that characteristics such as number of rows per ear, kernels per row, and ear diameter showed robust association with grain yield.

Molecular markers are powerful tools for assessing genetic diversity (*Bidyananda et al., 2024*; *Kamara et al., 2024a*). To comprehensively assess the genetic diversity among 14 maize inbred lines, SCoT, CDDP, and SRAP molecular markers were employed. SRAPs target coding regions, offering insights into functional diversity relevant to phenotypic traits like arid adaptation (*Li & Quiros, 2001*). SCoT markers also focus on genic regions, providing another perspective on functional diversity (*Collard & Mackill, 2009*). CDDP markers reveal polymorphism in both coding and non-coding regions, offering a broader view of genetic architecture (*Aouadi et al., 2019*). While each marker has limitations, their complementary strengths allowed for a thorough, multi-dimensional understanding of genetic diversity, which was prioritized at this stage of germplasm evaluation. The assessment of genetic diversity revealed substantial variation within the germplasm. The high level of polymorphism observed, particularly with SCoT markers, indicates a diverse genetic background among the inbred lines. These findings are consistent with previous studies, which have demonstrated the effectiveness of these marker systems in uncovering genetic variation in maize (*Brunner et al., 2005*; *Prasanna, 2012*). By integrating multiple marker systems with advanced sequencing techniques, researchers can gain comprehensive insights into plant genetic resources and support conservation efforts. The genetic distance analysis, based on the Dice coefficient, further highlights the differentiation among the inbred lines. The clustering patterns observed in the dendrogram (Fig. 10) suggest the presence of distinct genetic groups. This information can be valuable for breeders in selecting diverse parents for hybridization and developing new cultivars with improved agronomic traits. Identifying closely related lines, such as LZAM7B and ZBM40A, can be useful for marker-assisted selection (MAS) programs. By developing molecular markers linked to specific traits of interest, breeders can efficiently select for these traits in early generations. Moreover, identifying genetically distant lines, such as LZAM7B and DKCA2, can facilitate the creation of diverse breeding populations and enhance genetic recombination.

Molecular data, particularly from SCoT, CDDP, and SRAP markers, provided a comprehensive view of the genetic diversity present among the inbred lines, which is crucial for identifying genetic markers associated with resilience to environmental stress. By integrating molecular markers with agronomic data, breeders can enhance the selection process, targeting traits that confer resilience to environmental stresses. In light of the

increasing challenges posed by climate change, particularly in arid and semi-arid regions, this study indicates the potential of combining molecular techniques and phenotypic evaluation to breed maize hybrids tolerant to environmental stresses. The incorporation of molecular markers into maize breeding strategies could accelerate the development of maize hybrids with improved tolerance to environmental stress, ultimately contributing to food security in regions vulnerable to water scarcity and changing climate conditions.

## CONCLUSIONS

Integrating field evaluation and molecular marker techniques for fourteen Egyptian maize inbred lines unveiled substantial genetic diversity and phenotypic variation. The field evaluation of maize lines over three growing seasons revealed significant variability in phenological traits, plant stature, ear characteristics, and grain yield. This diversity indicated genetic potential within the assessed germplasm and its value for maize breeding programs. Early-maturing lines like IKA22 demonstrated suitability for short growing seasons, while high-yielding lines RA28C, ZBM40A, DKCA2, and LZAM7B were identified as promising candidates for hybrid development. Cluster analysis classified assessed lines based on yield traits, identifying high-performing clusters (*e.g.*, Group A with RA28C, and Group B with DKCA2, ZBM40A, and LZAM7B) and low-yielding clusters (*e.g.*, Group F with IKA22) that offer prospects for targeted improvement. Principal component analysis and heatmap clustering further validated these findings, demonstrating robust correlations among traits and genetic divergence among lines. The strong associations between grain yield and critical characteristics, including ear diameter, number of kernels/row, and rows per ear, highlight their importance as selection criteria for improving productivity. The molecular analysis provided valuable insights into genetic diversity and associations among the evaluated inbred lines. The integrated phenotypic and molecular characterization of these Egyptian maize inbred lines provides a valuable resource for developing climate-resilient maize cultivars adapted to arid conditions. By strategically utilizing the identified superior traits and managing genetic diversity, breeding programs can effectively address the challenges of water scarcity and contribute to sustainable maize production in these environments.

### Funding

This work was funded by the Princess Nourah bint Abdulrahman University Researchers Supporting Project number (PNURSP2025R366), Princess Nourah bint Abdulrahman University, Riyadh, Saudi Arabia and the Deanship of Scientific Research and Graduate Studies at King Khalid University through a large group Research Project under grant number RGP2/342/45. The funders had no role in study design, data collection and analysis, decision to publish, or preparation of the manuscript.

### Grant Disclosures

The following grant information was disclosed by the authors:

The Princess Nourah bint Abdulrahman University Researchers Supporting Project number, Princess Nourah bint Abdulrahman University, Riyadh, Saudi Arabia: PNURSP2025R366.
The Deanship of Scientific Research and Graduate Studies at King Khalid University through a large group Research Project: RGP2/342/45.

## Competing Interests

Diaa Abd El-Moneim is an Academic Editor for PeerJ.

## Author Contributions

- Abdallah A. Hassanin conceived and designed the experiments, performed the experiments, analyzed the data, prepared figures and/or tables, authored or reviewed drafts of the article, and approved the final draft.
- Areej S. Jalal conceived and designed the experiments, performed the experiments, analyzed the data, prepared figures and/or tables, authored or reviewed drafts of the article, and approved the final draft.
- Ehab M. Mahdy conceived and designed the experiments, performed the experiments, analyzed the data, prepared figures and/or tables, authored or reviewed drafts of the article, and approved the final draft.
- Hend Mandour conceived and designed the experiments, performed the experiments, analyzed the data, prepared figures and/or tables, authored or reviewed drafts of the article, and approved the final draft.
- Elsayed Mansour conceived and designed the experiments, performed the experiments, analyzed the data, prepared figures and/or tables, authored or reviewed drafts of the article, and approved the final draft.
- Mohamed M. Kamara conceived and designed the experiments, performed the experiments, analyzed the data, prepared figures and/or tables, authored or reviewed drafts of the article, and approved the final draft.
- Mohammed O. Alshaharni conceived and designed the experiments, performed the experiments, analyzed the data, prepared figures and/or tables, authored or reviewed drafts of the article, and approved the final draft.
- Eman Fayad conceived and designed the experiments, performed the experiments, analyzed the data, prepared figures and/or tables, authored or reviewed drafts of the article, and approved the final draft.
- Mohammed Alqurashi conceived and designed the experiments, performed the experiments, analyzed the data, prepared figures and/or tables, authored or reviewed drafts of the article, and approved the final draft.
- Maha Aljabri conceived and designed the experiments, performed the experiments, analyzed the data, prepared figures and/or tables, authored or reviewed drafts of the article, and approved the final draft.
- Thorya Abdulrahman Fallatah conceived and designed the experiments, performed the experiments, analyzed the data, prepared figures and/or tables, authored or reviewed drafts of the article, and approved the final draft.

- Diaa Abd El-Moneim conceived and designed the experiments, performed the experiments, analyzed the data, prepared figures and/or tables, authored or reviewed drafts of the article, and approved the final draft.
- Ahmed S. Eldomiaty conceived and designed the experiments, performed the experiments, analyzed the data, prepared figures and/or tables, authored or reviewed drafts of the article, and approved the final draft.

## Data Availability

The raw data is available in the Supplemental Files.

## Supplemental Information

Supplemental information for this article can be found online at http://dx.doi.org/10.7717/peerj.19598#supplemental-information.

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
