# Peer review of "Unravelling agronomic performance and genetic diversity of newly developed maize inbred lines for arid conditions"

_PeerJ, doi:10.7717/peerj.19598_

## Round 0.1 · original submission · Major Revisions

Especially if you cannot address the concerns of reviewer 5, my next decision may be to reject your manuscript.

Reviewer 1 ·

Basic reporting

The manuscript is well-written, with clear and professional English used throughout. The structure of the article is professional, with appropriate sections and sub-sections. The figures and tables are well-presented and support the findings effectively. The article is self-contained, with relevant results that address the hypotheses and research questions posed. However, there are a few areas where improvements could enhance the clarity and completeness of the manuscript:
1- Environmental Context: Include more detailed descriptions of the environmental conditions during the three growing seasons, such as soil moisture levels, irrigation practices, and any biotic stresses encountered. This information is crucial for understanding the phenotypic expression of the traits measured.
2- Inbred Line Selection: Clarify the criteria used for selecting the 14 inbred lines for evaluation. Were they chosen based on prior performance, genetic diversity, or other criteria? This will help readers understand the rationale behind the selection process.
These minor improvements will enhance the overall quality of the manuscript, ensuring it meets the high standards expected for publication.

Experimental design

The experimental design of the study is robust and well-structured, with a clear description of the field trials, molecular analyses, and statistical methods used. The use of a randomized complete block design with three replicates over three growing seasons is appropriate for assessing the stability and performance of the inbred lines under arid conditions. The integration of molecular marker techniques (SCoT, CDDP, and SRAP) to evaluate genetic diversity adds significant value to the study.

Validity of the findings

The findings of the study are valid and well-supported by the data presented. The use of multi-season field evaluations and molecular marker analyses provides a comprehensive assessment of the genetic diversity and agronomic performance of the maize inbred lines. The statistical analyses, including principal component analysis (PCA) and clustering, are appropriate and effectively support the conclusions drawn.

Additional comments

The manuscript presents a well-executed study that integrates field evaluations and molecular marker analyses to assess the genetic diversity and agronomic performance of newly developed maize inbred lines under arid conditions. The findings are significant and have clear implications for maize breeding programs aimed at improving yield and resilience in challenging environments.

Reviewer 2 ·

Basic reporting

.

Experimental design

.

Validity of the findings

This study effectively integrates phenotypic and molecular analyses to assess the genetic diversity of Egyptian maize inbred lines, providing valuable data for targeted breeding strategies aimed at enhancing yield and resilience in arid conditions.
-Line 34: add full stop after considerably to be (Plant and ear heights varied considerably. Taller lines (such as Gz603 and SD34) potentially offered greater photosynthetic capacity, while shorter lines (such as Gm2, SD63, and Gm4) may have exhibited improved lodging resistance, especially under adverse weather conditions.)
- Line 40: Change "showed" to "showing" to match the grammar of the rest of the sentence. "Grain yield per plant also revealed seasonal variation, with SD34, SK13, Gz628, and SK12 showing higher yields."
- The introduction effectively establishes the importance of maize as a global staple and the challenges facing its production.
- Line 71 remove “Moreover”
- Line 72replace especially with particularly
- The sentence in lines 81,82 could be shortened to "Identifying inbred lines with superior agronomic traits is essential for developing high-yielding, adaptable hybrids."
- In statistical analysis, it is not specified which R packages were used (is it version 4.2 2?).
- The ethidium bromide concentration is given as 0.5 μg/ml (Please double check this concentration)
- Revise The sentence in lines 182-184 to (Figure 1A illustrates the days to tasseling for each inbred line across the three growing seasons. Gm2 was the earliest maturing line across all seasons. In the first season, Gm2 was followed by Gz628, SK12, SD63, SK5, and SK9, while in the second and third seasons, it was followed by SK5, SK13, SK9, and SD34.)
- Revise The sentence in lines 280-281 to (Figure 5 shows the heatmap based on the evaluated agronomic traits, revealing distinct clustering of maize lines. The blue color represents the highest trait values, and the red color represents the lowest trait values.)

Reviewer 3 ·

Basic reporting

The article is written in English and is easy to understand. It contains enough literature to support and compare the findings. However, the figures showing the agronomic performance of inbred maize lines are confusing and may be reorganized.

Experimental design

Drought is a global phenomenon, and developing new varieties or progenies is crucial for the sustainability of the food supply. This study aims to determine drought-resistant maize inbred lines through genetic analysis. The experimental design of the study is suitable for achieving the target. However, it is not clear whether irrigation was applied.

Validity of the findings

The findings are attractive for readers and valid for breeders.

·

Basic reporting

I appreciate the invitation to review the manuscript entitled "Unravelling agronomic performance and genetic diversity of newly developed maize inbred lines for arid conditions". The manuscript provides a comprehensive study of the agronomic performance and genetic diversity of newly developed maize inbred lines in arid conditions, making a valuable contribution to maize breeding and genetic improvement. The combination of field evaluation and molecular analysis across three growing seasons adds robustness to the findings. The study highlights key traits such as phenological attributes, plant stature, ear characteristics, and grain yield, which are crucial for improving maize resilience. However, major improvements are required.

Experimental design

- Provide clearer connections between the importance of genetic diversity, maize breeding, and the specific challenges under arid conditions. Highlight the gaps the study addresses. The rationale for using the specific molecular marker techniques (SCoT, CDDP, and SRAP) should be elaborated further.
- More explanation of the experimental conditions is required.

Validity of the findings

- In results section, due to the extensive data, consider splitting complex sections into smaller sub-sections with clearer headings. Seasonal variation in traits is an important point. However, the discussion of these variations could be more structured to better highlight the implications for breeding.
- In results section, provide a clearer linkage between the agronomic performance data (phenological traits, plant stature, etc.) and their practical significance in breeding programs.
- The PCA and heatmap are useful, but they need better integration into the discussion to highlight their relevance to the research question.
- Legond of figure 8 should be “Phylogenetic tree of 14 Egyptian maize inbred lines was revealed based on specific SCoT, CDDP, and SRAP marker data.”
- Please correct the Caption of Table 2 to SCoT, CDDP and SRAP primers with their nucleotide sequences, molecular weight g/mol, primer length, and GC % content
- In table 2 please add (nt) after primer length.

Additional comments

- Please revise the abstract to focus on the primary findings, such as the identification of superior inbred lines and the usefulness of molecular markers for genetic diversity analysis.
- In discussion section, explain how the identified inbred lines can be utilized to develop high-yielding, resilient hybrids suitable for arid conditions.
- The seasonal fluctuations should be explored in terms of environmental impact. Elaborate on how environmental conditions affected trait expression across seasons. Discuss how the molecular data can inform breeding strategies, especially in arid environments. Comparing with more recent research on climate resilience and maize breeding under drought stress would strengthen the discussion.
- In conclusions section, provide how the identified traits and genetic diversity findings can guide breeding programs targeting arid conditions.

Reviewer 5 ·

Basic reporting

First of all, as a result of my personal evaluation; it can be said that the English, literature and article structure used throughout the article are appropriate.
However, I would like to share the opinion that the article does not meet the journal's standards due to uncertainties in the plant material and molecular techniques used.
However, I would like to share the opinion that the article does not meet the journal's standards due to uncertainties in the plant material and molecular techniques used. These two very important issues can be explained as follows.
1) There is uncertainty as to why the 3 molecular techniques were chosen. The reasons for their selection should have been explained well. Would it be sufficient to choose the most reliable one that would serve your purpose?
2) It is not clear whether the inbred lines are truly of different genetic origin. Moreover, even if this is the case, it is clearly not an approach that will determine their use in hybrid variety breeding. Because in hybrid corn breeding, in addition to the variability status of the inbred lines, their general and specific combining abilities will be more useful according to their performance among their hybrids. Because it is a known fact that genotypes with high variability are not a prerequisite for high-performance hybrid varieties.

Experimental design

No comment

Validity of the findings

No comment

Additional comments

It would be more appropriate to publish this article in a regional journal.

Annotated reviews are not available for download in order to protect the identity of reviewers who chose to remain anonymous.

---

## Round 0.2 · accepted · Accept

The version of the manuscript after corrections is suitable for publication. Correct the following minor errors at the proofing stage.

Line 418 "Long et al. [45]," incorrect citation. Correct please.

Line 506-508 you mention analysis of combining abilities but there is no data on this. Delete these lines. It is confusing.

Reviewer 1 ·

Basic reporting

The manuscript fulfills all the listed requirements and meets the standards expected for publication

Experimental design

The manuscript meets the required criteria for originality, relevance, methodological rigor, and transparency

Validity of the findings

The manuscript also meets the requirements for transparency, methodological soundness, and responsible scientific reporting

Reviewer 2 ·

Basic reporting

The authors addressed all comments. Thank you.

Experimental design

The authors addressed all comments. Thank you.

Validity of the findings

The authors addressed all comments. Thank you.

Additional comments

The authors addressed all comments. Thank you.

·

Basic reporting

thanks for considering all my previous comments. The manuscript was improved accordingly and now is suitable for publication

Experimental design

Research question well defined, relevant & meaningful. It is stated how research fills an identified knowledge gap. the Experimental design was suitable to test their questions and hypothesis.

Validity of the findings

All underlying data have been provided; they are robust, statistically sound, & controlled.
Conclusions are well stated, linked to original research question & limited to supporting results.

Additional comments

The authors considered all my previous comments. The manuscript was improved accordingly and now is suitable for publication